# Exploring the evidence of direct threats to cetaceans from maritime vessels: A systematic map

Valeria Ferrari[1]*, Emily Hague[1], Marija Sciberras[2], Karen A. Alexander[3], Patrick D. O'Hara[4,5], Lauren McWhinnie[1,5]

1 Institute of Life and Earth Sciences, School of Energy, Geoscience, Infrastructure and Society, Heriot-Watt University, United Kingdom, 2 The Lyell Centre, School of Energy, Geoscience, Infrastructure and Society, Heriot-Watt University, United Kingdom, 3 International Centre for Island Technology, Institute of Life and Earth Sciences, School of Energy, Geoscience, Infrastructure and Society, Heriot-Watt University, United Kingdom, 4 Integrated Marine Spatial Ecology Lab, Institute of Ocean Sciences, Environment and Climate Change, Canada, 5 Department of Geography, University of Victoria, Victoria, British Columbia, Canada

* vf2001@hw.ac.uk

## Abstract

Cetaceans face a multitude of well-recognised anthropogenic threats, many of which can be attributed to the activities of marine vessels that are increasing in number throughout the world's oceans. This study applies a systematic map methodology to better understand the current state of knowledge on vessel impacts to cetaceans, and to identify data gaps relating to specific geographies, vessel types and species. Literature searches were undertaken in January 2023 using three databases (Scopus, Web of Science, ProQuest), yielding 28,452 results. After duplicate removal and title, abstract and full-text screening, 568 documents were included in this review, resulting in 661 records of empirical evidence being extracted for further analysis. These records highlighted a focus on certain species (bottlenose dolphins (n = 133) and humpback whales (n = 89)) and vessel types (e.g., eco-tourism boats (n = 145)), and the majority of records were from North American waters (n = 274). There was also limited evidence demonstrating impacts of vessels for entire groups of species including porpoises (n = 21) and beaked whales (n = 22). Given the global distribution of marine mammals and vessels, there were few published records available for African waters and international waters. However, for 41.4% of the records it was not possible to classify the type(s) of vessels represented. Therefore, greater clarity and recognition of the heterogeneity of vessels and their associated impacts would both help improve our understanding of potential knowledge gaps and, importantly, help refine our ability to holistically evaluate and assess the risk(s) maritime traffic poses to cetaceans.

**Data availability statement:** All relevant data are within the paper and its Supporting Information files.

**Funding:** The author(s) received no specific funding for this work.

**Competing interests:** The authors have declared that no competing interests exist.

## 1. Introduction

Maritime vessel traffic is now recognised as a pervasive and intensifying source of anthropogenic stressors across the world's oceans [1]. For the past few decades, marine transportation has undergone continuous increasing growth in terms of the number of vessels at sea, but also in relation to the size of vessels and their propulsion capabilities [2–4]. The International Chamber of Shipping documented that commercial shipping (at least 100 gross tons) now accounts for 105,500 vessels, amounting to around 85% of trade occurring globally [2]. This number is projected to increase in the future [2] as globalisation and demands for international trade continue to grow and more proposals for new shipping routes are being discussed [5,6]. Non-commercial traffic in coastal areas has also increased significantly in recent decades driven in part by coastal developments, maritime industries (e.g., fishing) and an increase in recreational boating and tourism [7,8]. Many coastal zones have undergone rapid economic expansion due to Blue Growth policies, resulting in a surge in vessel activity ranging from larger commercial cruise ships, to industrial operations and maintenance vessels, to small private boats [9–11]. This growth in maritime activity has also resulted in an increased awareness of the resultant pressures vessels can place on marine environments including various forms of noise, air and water pollution, as well as collision with marine megafauna [12–23]. The widespread and diverse nature of vessel activities means that these potential impacts are not just confined to inshore industrial regions but also extend to remote, vulnerable and/or protected areas. And this is only likely to increase further as Blue Economy industries move offshore. Furthermore, the potential environmental impacts of vessel traffic may be exacerbated by other phenomena such as climate change and biodiversity loss, or amplified due to cumulative or co-occurring stressors, leading to complex and often poorly understood ecological consequences [24–27].

Although often considered to be a more cost-efficient, safe and 'greener' form of travel and transportation [28,29], vessel activity can result in notable environmental impacts, such as increased noise, chemical and air pollution, introduction of non-native species, disturbance to seabeds and benthic communities, collision with marine megafauna (including penguins, turtles, pinnipeds, sharks, sirenians and cetaceans) [30–32]. Indeed, many pressures now recognised in marine and coastal habitats can either be directly or indirectly attributed to vessels [33,34]. A broad range of threats have been connected to maritime traffic, and subsequently documented for many marine species but cetaceans as a group have been identified as being particularly vulnerable to direct threats from the vessels themselves such as: underwater noise [13,35–44], discharges and contaminants [17,45,46], and vessel strikes [14–16,18,20–22,47–55]. In addition, there is a large body of evidence showing that activities associated with different vessel types such as pile driving, seismic surveys, sonar, dredging and fishing gear can also have a negative impact on cetaceans (e.g., displacements from core habitats, changes in vocalization, masking and entanglement) [35,56–64]. These 'secondary' impacts are often considered in environmental assessments, while impacts that are directly related to the vessels themselves can sometimes be overlooked as a contributing stressor [65–70].

While direct interaction between cetaceans and vessels may be localized, the wider ecological impacts are often experienced and measured elsewhere. For example, although vessel strikes occur at the specific location of the vessel [71], demographic changes, declines in population and ecosystem consequences are manifested and detected far from the site of collision [72–75]. And other vessel-related threats (like underwater noise generated by vessels) have a much broader spatial footprint, with the potential to affect cetaceans over greater distances, with the magnitude of responses depending on species sensitivity, habitat characteristics, and source levels [76–78]. For example, humpback whales reduced the number of vocalizations when vessel traffic passed within 1.2 km away from the animal [79], Blainville's beaked whales (*Mesoplodont densirostris*) altered their feeding behaviour in response to vessel noise up to distances of 5 km [35], and harbour porpoises (*Phocoena phocoena*) showed avoidance responses up to 4 km away from construction-related vessels [80].

These behavioural changes in response to increased noise levels have also been documented for repeated or prolonged periods of vessel exposure [81,82]. Individuals that are disturbed or have interactions with vessels will potentially alter their behaviour or have a physiological response to the encounter. Behavioural responses generally include changes in behavioural state (e.g., from resting or foraging to travelling; [83–87]), in swimming patterns (e.g., more "erratic" travelling direction, increased speed, changes in breathing intervals and diving behaviour; [88–95]) and in group cohesion [96,97]. Vessel noise has also been shown to result in changes in vocal behaviour including modification of the frequency range used and the rate of vocalization, with instances where animals even stopped vocalising altogether [39,98–104]. Introduction of vessel noise also has the potential to result in acoustic masking for some species that vocalise in similar frequency ranges as the vessels, which can affect their ability to communicate and navigate effectively [105–110]. Vessels can also have direct physical impacts: physiological responses have been documented during exposure to noise and disturbance [65,111–115], and a number of vessel types have been evidenced to cause both lethal and sub-lethal injuries to cetaceans as a result of vessel strikes [15,47,51,54,74,116–120].

Evaluating the potential severity of vessel impacts or risk they pose to cetaceans is complex. Cetaceans spend much of their time submerged, while many research approaches gather evidence of impacts from visible changes in above-water physical behaviour [12] or injuries [121], and tend to focus on acute, short-term effects that are generally more readily observed. Therefore we have less of a grasp of the long-term chronic impacts resulting from vessels [12]. Similarly, while there is increasing evidence of the impacts of vessels on individual cetaceans, the potential population-level implications are less well understood [122,123], though vessels have now been documented to contribute to declines in population size, changes in distribution, and even long-term displacement from key habitats for some species [32,100,124–133].

Growth in marine traffic has fundamentally led to an increase in the spatial and temporal overlap, or co-occurrence, between vessels and cetaceans, and consequently resulted in conservation challenges for several populations [134,135]. Those species that inhabit productive coastal areas and shelf seas would appear to be particularly vulnerable as they coincide with some of the world's busiest shipping corridors, fisheries, energy infrastructure and tourist areas [136–140]. However, the potential for chronic or acute impacts will vary depending on the stressor being considered, i.e., vessel noise can affect the recipient over different distances while strike risk is dependent on direct co-occurrence. The length of exposure to the stressor will also elevate the risk of impact and its potential severity, for example responses such as habitat displacement [141,142], behavioural change [60,143] and temporary (or permanent) threshold shifts [144] have all been shown to be influenced by the frequency and duration of exposure.

Recent advances in technologies such as passive acoustic monitoring (PAM) and remote sensing (Automatic Identification Systems (AIS) data) have advanced our understanding and ability to evaluate and in some cases mitigate the threats posed by vessels [15]. For many species and sea areas, improved data capture capabilities have increased our capacity to assess the risks posed by vessels, e.g., through calculating listening space reductions [145] or identifying areas of heightened collision risk [21,146]. This has given rise to many studies that model and predict vessel impacts, which can be used to support empirical studies or management and mitigation efforts.

Previous reviews have focussed on describing the effects of specific type of impact [147] or quantifying evidence available for a particular species or group of recipient species [12], vessel type or activity [12,32], or defined geographic

areas [148]. Few, however, have taken a holistic approach, assessing global literature on marine vessel impacts across all vessel types and cetacean species. This study aims to, for the first time, synthesise the currently available evidence for all marine cetaceans, all vessel types and all geographies, allowing for a fuller understanding of this issue. Given the large and highly varied body of literature on this topic, a systematic map was used, over a more extensive literature review, in order to better highlight knowledge gaps and quantify the spatial distribution of the evidence. In particular, with this study, we asked the following questions:

- What is the current global state of knowledge for impacts directly associated with maritime vessels on cetaceans?

- How does this state of knowledge vary across species and with vessel type?

## 2. Materials and methods

This study adopted a systematic map protocol, summarised below and guided by the Collaboration for Environmental Evidence Guidelines and Standards for Evidence Synthesis [149] and Preferred Reporting Items for Systematic Reviews and Meta-Analyses (PRISMA) protocol [150].

### 2.1. Search stage

Search terms were chosen to target marine cetacean species and different types of maritime vessels, and were combined into search strings, using appropriate database-specific syntax. Searches were conducted on 31st January 2023 in the following bibliographic databases: Scopus and Web of Science for peer-reviewed publications, and ProQuest for peer-reviewed and grey literature (search strings and number of documents found per database are provided in S1 Table in S1 Appendix).

### 2.2. Eligibility criteria

Only documents in English were included. Eligibility criteria were developed based on the PICO/PECO elements described below:

- Population: any wild, extant marine cetacean population (the full list of species is provided in S2 Table in S1 Appendix).

- Intervention/Exposure: immediate, mechanistic impacts that result from the physical presence, movement, or operation of a vessel without an intermediary process (direct impacts; e.g., ship strike, vessel noise, disturbance). Impacts arising as a consequence of vessel activity, but not from the vessel's immediate action (indirect/secondary impacts; e.g., air guns, fishing gear) as well as those pertaining to subsistence or commercial whaling were not considered eligible.

- Comparators: documents were not required to include a control or comparator.

- Outcomes: any impact on animal behaviour (such as vocalizations, behavioural state), distribution or physiological changes (such as hormone level).

In addition, only studies presenting new data were included (either empirical studies or reviews of previous data with new evidence provided). Both field-based observations or experiments (e.g., controlled exposure experiments) and modelled or simulated responses were considered eligible as long as an impact was assessed. Studies where spatial overlap between cetacean and vessel distribution is reported but no measure of impact is presented were excluded.

### 2.3. Screening stage

Results from all the bibliographic database searches were imported into the systematic review application "Rayyan" [151] for the screening process. The initial search resulted in a total of 28,452 documents (Fig 1). The "Duplicate Detection" function in Rayyan was used to identify duplicate records, which were then manually inspected for confirmation, resulting

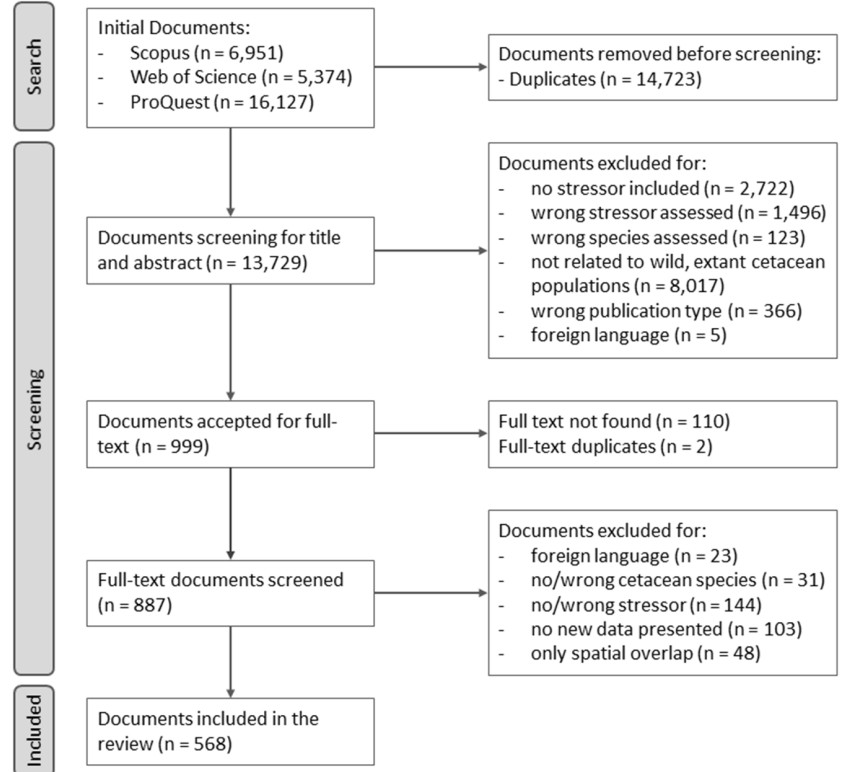

**Fig 1. PRISMA flow diagram detailing the volume of literature found, included and excluded during the search and screening stages of this systematic map.**

in a final 13,729 reports that underwent the screening process. Documents were first screened by title and abstract only, following the aforementioned eligibility criteria; uncertain documents were also forwarded to allow decision at the full-text screening stage. 999 documents were accepted for title-abstract screening. Of these, 110 could not be accessed or located, a further 2 were identified as duplicates and 319 were excluded after reading the full-text. This resulted in a final number of 568 documents included in this systematic map (Fig 1).

To test the clarity of the eligibility criteria, one reviewer screened all 13,729 abstracts, while a second reviewer screened only a subset (2,125, or 15%). The two processes were carried out independently and decisions were compared afterwards. There was a high rate of agreement between the two reviewers, with only 0.6% (n = 12) of conflicting decisions.

## 2.4. Data extraction and coding

All 568 included documents were manually processed by the main reviewer to compile the relevant information into a single Excel database. This included:

- Document-specific information (publication year, document type).

- Study location (latitude and longitude): whenever a map was provided in the document, the coordinates of the centre of the study area were recorded as location; otherwise, the study area and location were approximated based on the description provided. In case a study area covered multiple EEZs, a new record with a new location was added for each EEZ. Data on study locations was then aggregated into different marine regions (Fig 2) based on the continent where each EEZ is located.

- <u>Species of interest</u>: species was recorded to the lowest available taxonomic level (up to species level; the full list of species found is shown in S2 Table in <u>S1 Appendix</u>). Species were then grouped together by family (Balaenidae, Balaenopteridae, Eschrichtiidae, Neobalaenidae, Delphinidae, Monodontidae, Pontoporiidae, Phocoenidae, Physeteridae, Kogiidae, Ziphiidae) for the analysis and records that didn't contain enough information for classifiion were coded as "unspecified cetaceans".

- <u>Vessel type(s) assessed</u>: vessel categories and types were assigned based on the definitions in <u>Table 1</u>. If vessel type wasn't explicitly stated, expert judgement was employed to estimate the predominant or likely size, and therefore type, of the vessel involved.

- <u>Type of Response Documented</u>: we noted whether each data record presented a prediction of potential risk or response from the animals (coded as: predictive), as opposed to providing observations of recorded response(s) (coded as: empirical). We also recorded whether the document reported any response to the vessel(s).

For the purpose of this review, the publications that were screened (peer-reviewed paper, thesis, report, etc.) are hereafter referred to as "documents". From each document, a new "record" was extracted for each species-stressor-country combination addressed, meaning that a single document often resulted in multiple records.

## 3. Results

### 3.1. Evaluating evidence

From the 568 included documents, 919 records were extracted and summarized in this systematic map.

The majority of the evidence identified by this systematic map involved empirical records (n = 661, or 71.9%) rather than predictive studies (n = 258, or 28.1%), and this was the case for all species groupings and vessel categories with the exception of commercial cargo and passenger vessels, for which only 41% of the records were of empirical nature (S1 Fig in <u>S1 Appendix</u>). The majority of all the records (87.8%) also reported one (or more) response(s) to the impacts from marine vessels (S2 Fig in <u>S1 Appendix</u>).

The results hereafter will only present empirical records, the data for predictive records can be reviewed in the Supplementary Material (S7-9 Tables and S1,2,4,5 Figs in <u>S1 Appendix</u>). These are presented separately as they represent different types of evidence: empirical observations of a response as opposed to speculated, modelled outcomes.

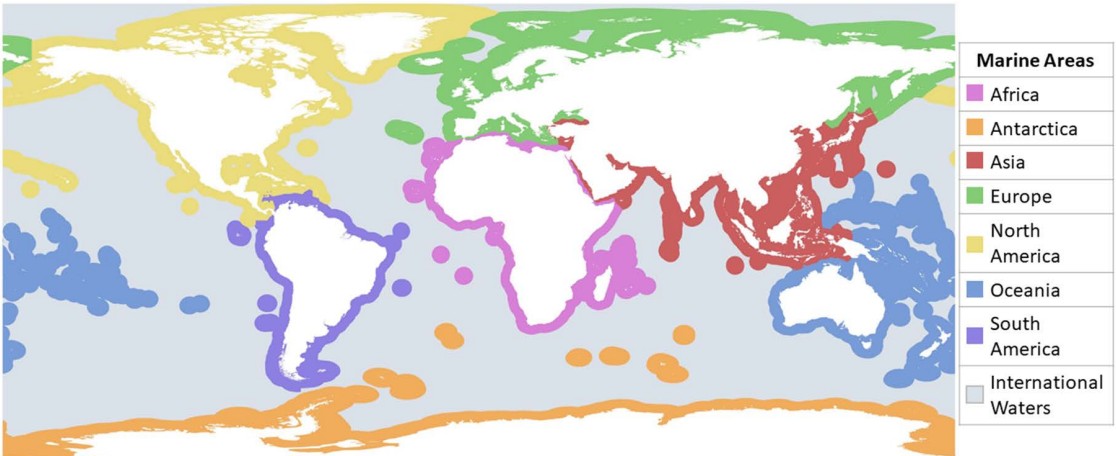

**Fig 2. Map of the different marine areas as defined for this systematic map.** World continent and EEZ layers obtained under CC BY 4.0 license from Natural Earth [152] and MarineRegions.org [153].

<u>https://doi.org/10.1371/journal.pone.0348502.g002</u>

**Table 1. List of the vessel types considered, as defined for the purposes of this systematic map.**

| Vessel Category | Vessel Type | | Definition (for the purpose of this analysis) |
|---|---|---|---|
| Commercial Cargo and Passenger (C&P) Vessels | Commercial AIS traffic | Any commercial AIS vessels | any vessel described as commercial non-passenger vessels (goods vessels and tankers) for which no further detail on vessel type was provided |
| | | Goods vessels | cargo, containers, bulk carriers, vehicle carriers |
| | | Tankers | any vessel transporting liquids or gasses |
| | Passenger AIS traffic | Any passenger AIS vessels | any vessel belonging to the AIS "Passenger" category (including, but not limited to, ferries, cruise ships and eco-tourism vessels) for which no further detail on vessel type was provided. |
| | | Cruise ships | Including small touring vessels (0–50 passengers), expedition ships (100–500 passengers) and international cruise (1000+passengers) |
| | | Ferries | any inshore, inter-island, coastal overnight, roll on roll off and passenger ferries of various sized |
| Support, survey, and government (SSG) vessels | Seismic vessels | | any vessel used for seismic surveys or geographical and geophysical activities |
| | Icebreakers | | any vessel specifically designed to carry out ice-breaking activities (and not any vessels with ice breaking capabilities) |
| | Cetacean research vessels | | any vessel used as dedicated platform for cetacean research* |
| | Supply vessels | | including support to offshore construction and aquaculture facilities |
| | Tugs | | all sizes |
| | Military vessels | | not including submarines |
| Fishing vessels | | | any vessel conducting fishing activities; sport and artisanal fishing vessels included |
| Recreational vessels | Motorboats | | small (<15m) boats with inboard or outboard engine; recreational motorised craft |
| | Eco-tourism vessels | | any vessel used by eco-tourism companies |
| | Jet-ski | | |
| | Sailing vessels | | monohulls, catamarans, trimarans |
| Unspecified vessels | Small unspecified vessels | | Likely small (<15-20m) vessels of unspecified type |
| | Large unspecified vessels | | Likely large (>20m) vessels of unspecified type, including AIS vessels that couldn't be assigned to any of the previous categories. |
| | Other unspecified vessels | | any case in which no detail was given on the vessel type and vessel size could not be inferred |

*The research platform was categorised as "cetacean research vessels" only if it was a dedicated vessel (as opposed to opportunistic, e.g., eco-tours, ferries) and if the vessel was analysed for its research activity (e.g., approaches during tagging/biopsy operations) or behaviour (e.g., eco-tours vs research), but not if it was included as a proxy for general vessel presence.

Most (552 out of 661) empirical records analysed were sourced from peer-reviewed publications (Fig 3). The remaining 109 records comprised of Master's or PhD thesis (n = 63), technical reports (n = 26) and conference proceedings (n = 19). There was also a single magazine article that contributed to the evidence synthesised.

The earliest online record was published in 1981, however it should be noted that the availability of digital documents was notably low until the early 2000s, with an average of only 3–4 records per year before 2000, rising to about 15 per year in the following decade, and exceeding 30 annually after 2010 (Fig 3).

In terms of the spatial distribution of these records, the majority were documented within the Exclusive Economic Zones (EEZs) of North American countries (n = 274), followed by countries of the European continent (n = 113), Oceania (n = 81), Asia (n = 67), South America (n = 60), Africa (n = 29) and Antarctica (n = 4) (Fig 4). There were relatively few records that took place within international waters (n = 10). For the remaining 23 (3.4%) records, the exact location of the study could not be attributed to a particular geographic area.

### 3.2. Species-specific results

Empirical records of vessel impact(s) were found for all Balaenidae (4), Eschrichtiidae (1). Neobalaenidae (1), Monodontidae (2), Physeteridae (1) and Kogidae (2) species and the majority of Delphinidae (28 out of 36) and Balaenopteridae

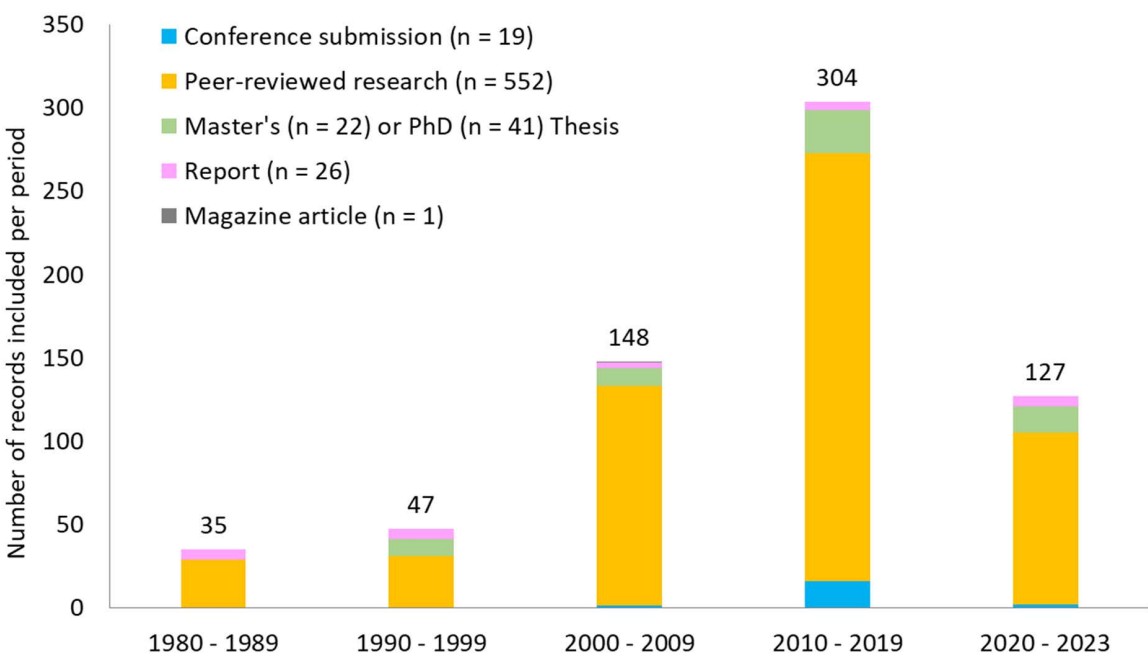

**Fig 3. Annual number of empirical records included in this systematic review, with focus on the document type.**

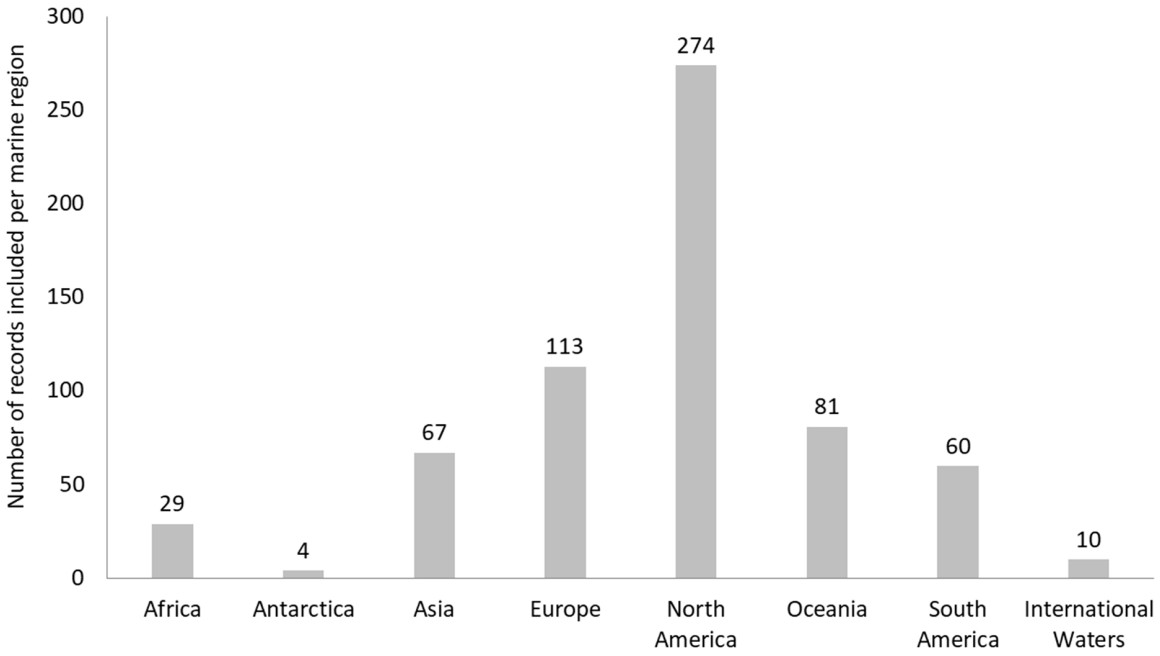

**Fig 4. Number of empirical records found for each of the marine areas identified in this systematic map.**

 

(8 out of 10) species. However, less than half of porpoises (Phocoenidae; 3 out of 7) and beaked whales (Ziphiidae; 7 out of 24) had any reported empirical evidence of vessel impacts, and no records were found for the Franciscana dolphin (family Pontoporidae) (S3 Fig in S1 Appendix).

The majority (46.4%) of records documenting vessel impacts involved species from the Delphinidae family (n = 307), followed by Balaenopteridae (n = 175), while much fewer records were found for the other families (less than 50 each) (S1a Fig in S1 Appendix). Only 1% (n = 8) of the records were more broadly classified as "cetaceans" due to the lack of further taxonomic details. Within each family, records were not evenly distributed between species (S3 Table in S1 Appendix). For the Balaenidae and Balaenopteridae families most records of vessel impacts involved, respectively, Northern Atlantic (*Eubalaena glacialis*; 17 out of 46) and Southern Right whales (*E. australis*; 18 out of 46), and humpback whales (*Megaptera novaeangliae*; 89 out of 175), while for Delphinidae the majority of records were attributed to bottlenose dolphins (*Tursiops* sp., 133 out of 307) and killer whales (*Orcinus orca*; 24 out of 307). For the family Monodontidae, records on beluga (*Delphinapterus leucas*; 15 out of 18) were 5 times more abundant than on narwhals (*Monodon monoceros*; 3 out of 18), and similarly, for the Kogiidae, pygmy sperm whales (*Kogia breviceps*; 6 out of 8) held triple the number of records than dwarf sperm whales (*K. sima*). Porpoise (Phocoenidae) records were dominated by a single species: harbour porpoise (*Phocoena phocoena*; 18 out of 21). Beaked whale (Ziphiidae) records on vessel impacts were limited, with the most evidence obtained for goose-beaked whales (*Ziphius cavirostris*; 9 out of 22) and only 1 or 2 for the other six species for which records were found.

The number of records attributed to different families over time reflects the same publication trends shown in Fig 3, with fewer online documents prior to 2000 and a notable increase after 2010 (S4a Table in S1 Appendix). By contrast, the number of species studied showed a more gradual increase for most families (S4c Table in S1 Appendix). Spatial trends in records are more challenging to interpret, for reasons discussed later. However, it is possible to discern that evidence of vessel impacts on baleen whales (Balaenidae, Balaenopteridae, Eschrichtiidae, Neobalaenidae), delphinids (Delphinidae and Monodontidae) and Physeteridae has been documented in all (or most) marine areas covered by the distribution ranges of those species. Records for Kogiidae was available only for Europe, North America and Asia, while there were no records of empirical evidence of vessel impacts on porpoises (Phocoenidae) and beaked whales (Ziphiidae) from South America, Oceania, Africa, Antarctica and international waters (S4b Table in S1 Appendix).

### 3.3. Vessel-specific results

Almost half (317 out of 661) did not specify the vessel type involved. For the remaining 344 records, recreational vessels were most abundant (n = 209), followed by support survey and government (SSG) vessels (n = 55), commercial cargo and passenger (C&P) vessels (n = 41) and fishing vessels (n = 39) (S1b Fig in S1 Appendix).

**3.3.1. Unspecified vessels.** Records indicating impacts from unspecified vessel types were found from all decades and steadily increased after 2000, to the point of taking up 59% of empirical records in the period between 2020 and 2023 (S5a Table in S1 Appendix). Records with unspecified vessel types are distributed across all continents, particularly in North America (152 out of 274) and Europe (68 out of 120) (S5b Table in S1 Appendix).

For 220 of the records where no vessel type was specified, no assumption could be made about the size of the vessel involved, due to the data being related to stranding events or scarring on the animals' body (n = 152), or from being obtained from locations where vessels of various sizes and types were likely to have been operating (n = 68). Of the remaining evidence, the majority involved large vessels (n = 61) as opposed to small ones (n = 36).

Records from large unspecified ships and other unclassified vessels focused mainly on impacts to Balaenopteridae species (n = 27 and 66, respectively) and those from small unspecified vessels were mostly related to Delphinidae species (n = 28), while comparatively few documented impacts on Physeteridae and Kogiidae sperm whales, beaked whales and porpoises (respectively n = 17, 7, 14 and 11 across all unspecified vessel categories) (Fig 5). In terms of species specifics, the records predominantly involved humpback whales (n = 44), bottlenose dolphins (n = 42), and sperm whales (*Physeter macrocephalus*, n = 17) (S6 Table in S1 Appendix).

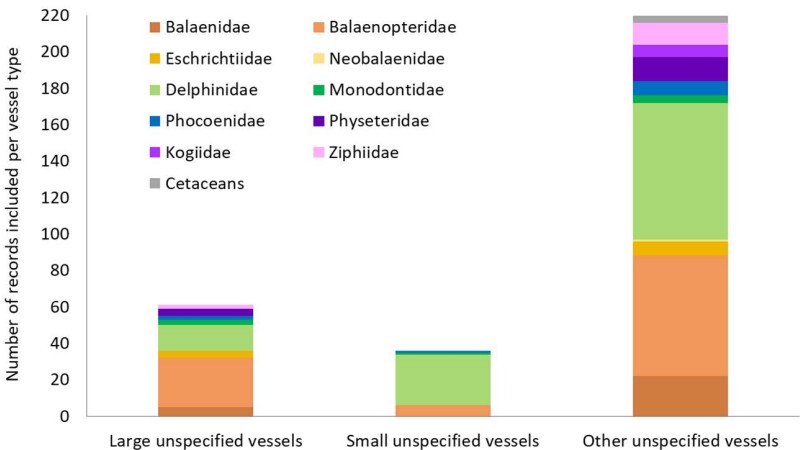

**Fig 5. Number of empirical records documenting the impacts of unspecified vessels on the different families of cetaceans.**

**3.3.2. Recreational vessels.** A substantial number of records related to recreational vessels were published after 2010, with a minimal number (n = 9) published before 2000 (S5a Table in S1 Appendix). Among these records, 69.4% (n = 145) corresponded to eco-tourism vessels, 24.8% (n = 52) to motorboats, 4.8% (n = 10) to sailing vessels, and only 1% (n = 2) to jet skis (Fig 6). Impacts from motorboats and eco-tourism vessels have been documented in all marine areas except Antarctica. Conversely, records focusing specifically on jet ski impacts have been found only in North America and Oceania (S5b Table in S1 Appendix).

Overall, recreational vessels have been primarily studied in relation to Delphinidae (n = 125) and Balaenopteridae (n = 47) species, while much less evidence was available for Phocoenidae, Kogiidae and Ziphiidae (two records each family) (Fig 6). However, the distribution of records varies across different vessel types (S6 Table in S1 Appendix). Eco-tourism vessels were associated with all cetacean families except porpoises and Neobalaenidae, but in particular

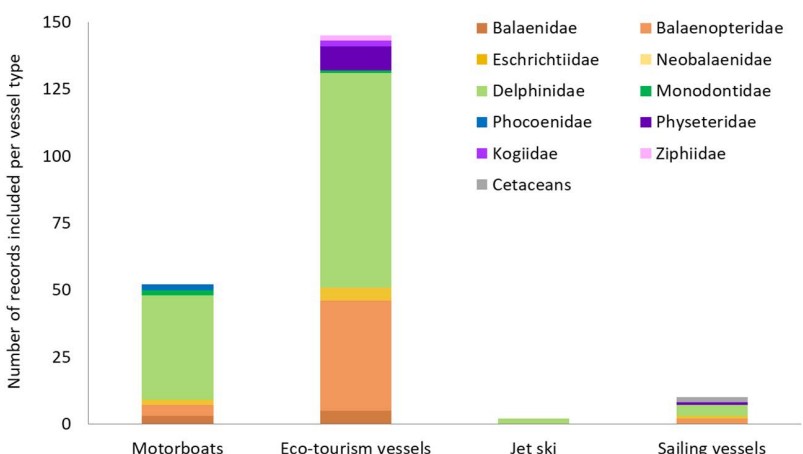

**Fig 6. Number of empirical records documenting the impacts of recreational vessels on the different families of cetaceans.**

bottlenose dolphins (n = 40) and humpback whales (n = 30). By contrast, impacts from recreational motorboats have only been recorded for baleen whales (Balaenidae, Balaenopteridae and Eschrichtiidae), delphinids (Delphinidae and Mono-dontidae), and porpoises (Phocoenidae), and sailing vessels only for Balaenopteridae, Eschrichtiidae, Delphinidae and Physeteridae. Published records evidencing impacts from jet-ski were limited (n = 2): a collision event with a common dolphin (*Delphinus delphis*) [154] and avoidance responses from bottlenose dolphins [155] (S6 Table in S1 Appendix).

### 3.3.3. Support, survey and government vessels.

The temporal distribution of records varies across SSG vessel types: for most, evidence of impacts only begins to emerge from the year 2000 onwards, with the exception of icebreaker vessels, whose records are concentrated in the 1980s and 1990s (S5a Table in S1 Appendix). Impacts from cetacean research vessels have been studied in most sea areas, however evidence for other SSG vessel types was typically limited to only one or two regions, with a notable absence of records from African waters (S5b Table in S1 Appendix).

61.8% (34 out of 55) of SSG records involved cetacean research vessels, while a far smaller proportion provided evidence from seismic and icebreaker vessels (n = 7 each), supply vessels (n = 6), and tugs (n = 1) (Fig 7).

Impacts attributed to SSG vessels have predominantly been documented for Delphinidae species (n = 25), with far fewer records for the other families (less than 10 each) (Fig 7). Despite the notable diversity of vessel types included in this category, there were relatively few records (n = 55) that reported interactions between SSG vessels and cetaceans. Often there was only one record evidencing impacts from a specific vessel type within this category on a singular species (e.g., effects of supply vessels on harbour porpoises), with some notable exceptions (S6 Table in S1 Appendix):

- disturbance from seismic vessels on bowhead (*Balaena mysticetus*; n = 3 [156–158]) and humpback whales (n = 4 [159–162]);

- effects on noise from icebreakers on beluga (n = 3 [163,164]);

- disturbance from cetacean survey vessels on bottlenose dolphins (4 [165–168]) and sperm whale (n = 6 [165,169–172]).

### 3.3.4. Commercial cargo and passenger vessels.

Evidence of cetacean impacts from commercial cargo and passenger vessels is greatest between 2010–2019, with only four records published before 2000 (S5a Table in S1 Appendix). Records for vessels specifically identified as commercial goods vessels and ferries could be found

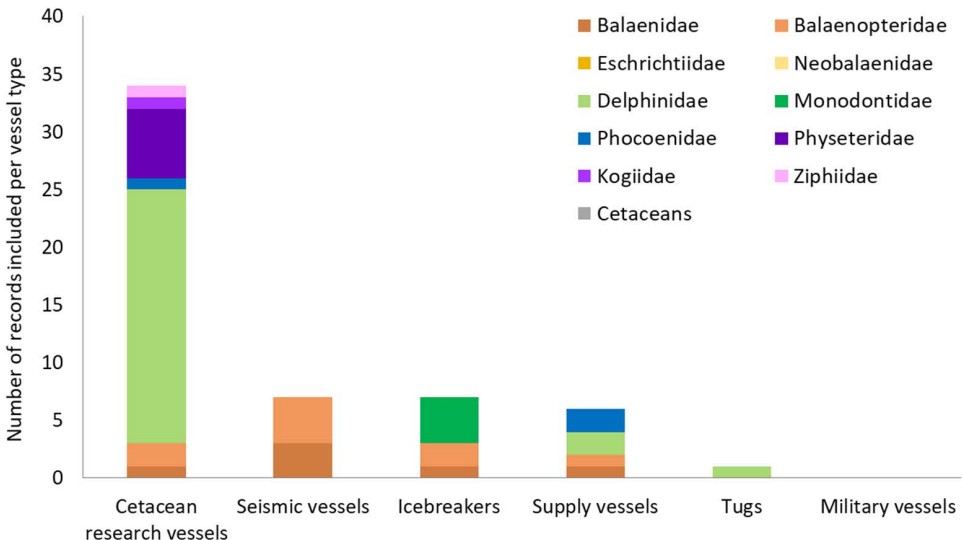

**Fig 7. Number of empirical records documenting the impacts of support, survey and government vessels on the different families of cetaceans.**

across most sea areas, while evidence of impacts for tankers and cruise ships was restricted to Europe and North and South America (S5b Table in S1 Appendix). It is important to note that some of these vessels were likely captured as part of the "large unspecified" and "other unspecified" vessels (mentioned in section 3.3.1.).

63.4% (26 out of 41) of the C&P vessels records referred to cargo traffic (in particular goods vessels, n = 16), while the remaining 36.6% (15 out of 41) was passenger traffic (notably ferries, n = 11) (Fig 8).

Most records involving C&P vessels considered their impacts on Balaenopteridae species (n = 16), followed by the Delphinidae (n = 15) (Fig 8), and these records were dominated by three species: bottlenose dolphins (n = 7), blue whales (*Balaenoptera musculus*; n = 5) and humpback whales (n = 4) (S6 Table in S1 Appendix). This review identified no specific records of C&P vessels directly impacting sperm whales (Physeteridae and Kogiidae) and only four records for beaked whales (Ziphiidae) and porpoises (Phocoenidae) (Fig 8). The only beaked whale record evidences a collision between a ferry (hydrofoil) and a Stejneger's beaked whale (*Mesoplodon stejnegeri*) in Japan [173], while the three porpoise records all involved harbour porpoises responding to noise or disturbance from commercial cargo vessels or ferries [174–176] (S6 Table in S1 Appendix).

### 3.3.5. Fishing vessels.
Before 2000, there were no records related to the impacts of fishing vessels (S5a Table in S1 Appendix). Evidence of impacts is available for all marine areas except Antarctica, with most records originating from North America (n = 12), Europe (n = 9), and Asia (n = 7) (S5b Table in S1 Appendix).

Most records (25 out of 39) associated with fishing vessels involved almost exclusively Delphinidae species, particularly bottlenose dolphins (Fig 9; S6 Table in S1 Appendix). Only 7 records were found for baleen whales (3 for humpback, 2 for southern right, 1 for minke whales (*Balaenoptera acutorostrata*) and 1 for an unidentified baleen species), while minimal evidence was available for porpoises (2 records on harbour porpoise) and beaked whales (2 records on goose-beaked whale and 2 for an unidentified beaked whale species). No impacts involving fishing vessels were reported for Physeteridae or Kogiidae species.

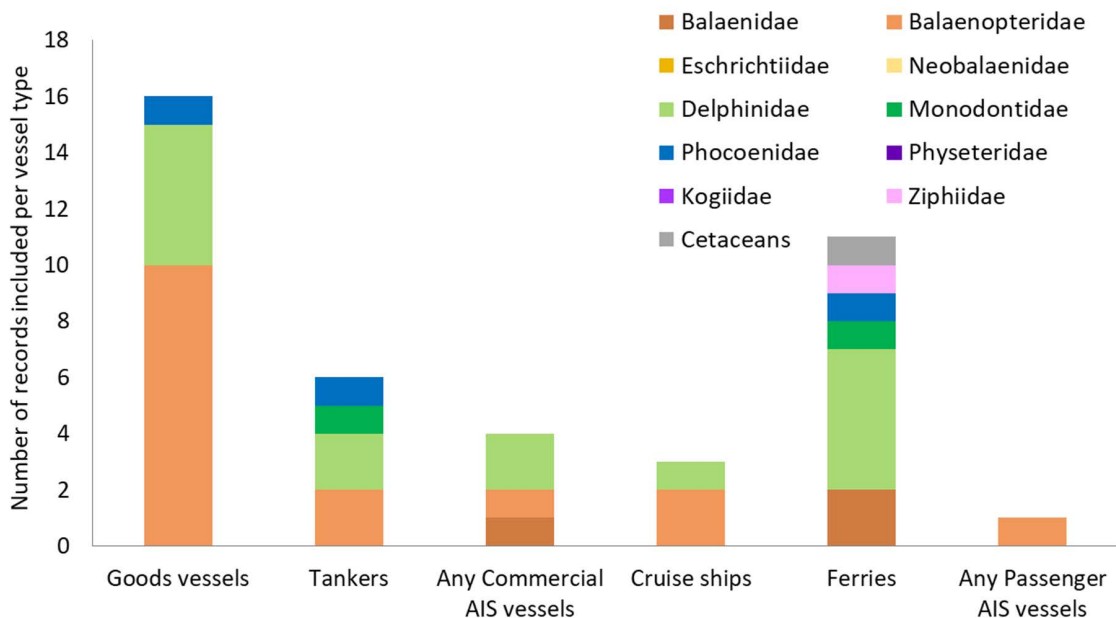

**Fig 8. Number of empirical records documenting the impacts of commercial cargo and passenger vessels on the different families of cetaceans.**

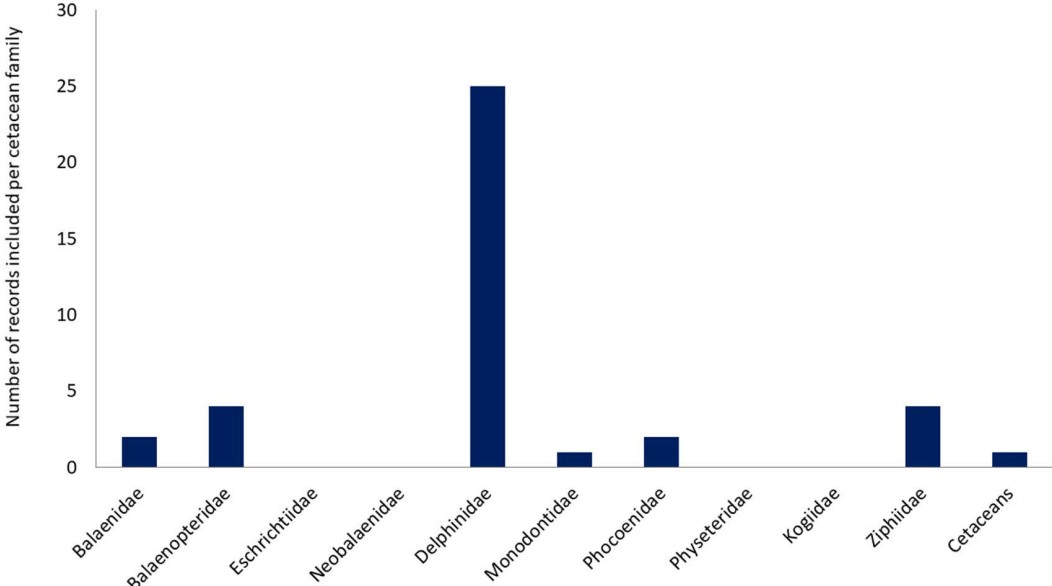

**Fig 9. Number of empirical records documenting the impacts of fishing vessels on the different families of cetaceans.**

## 4. Discussion

This systematic map highlights that, despite a significant volume of publications that consider maritime vessels and cetaceans, a comparatively smaller proportion (50%) of the literature reports on the direct impacts (e.g., vessel strike or disturbance associated with vessel presence or approach) that different vessel types can have on different groups of cetaceans. Importantly, the published knowledge identified in this review is unevenly distributed across species and vessel type(s). Despite this, a number of important findings have been identified.

The results show an increasing amount of published evidence over the decades, in particular after 2000. This is likely due to a growing interest in charismatic megafauna and awareness of the potential effects that anthropogenic activities as a whole can have on the environment [177], as well as higher availability of online digital documents and reduced barriers in the publication process [178,179].

Spatial trends are more difficult to infer. Larger regions may hold a higher number of records partly because of their size, and not necessarily only because of differences in research effort. Similarly, vessels may be unevenly distributed between marine regions, but accurate estimates on the number and types of vessels operating in each area are lacking, as the current reporting system appears to focus more on the country of registration rather than place of operation [2]. However, information on cetacean presence is more readily available and numbers are indeed comparable between the marine regions as defined for this systematic map (45-54 species, [180]). The results can therefore help shed light on possible geographical disparities in the research effort on vessel impacts, with most of the evidence being gathered in North American or European waters and far fewer records documenting impacts in international waters or other marine areas. This could be partially explained by the fact that this study reviewed only documents written in English: certain document types (like technical reports, thesis or grey literature) will likely be published in the country's main language and not always translated into English, which may have resulted in records from non-English speaking areas being under-represented in the results. Other factors that may have played a role in the geographical differences in the evidence base include the dominance of the Western scientific paradigm in research, funding availability, policy and conservation obligations, as well as challenges associated with surveying expansive or remote areas [181]. The observed variability in research outputs could also reflect

a global disparity in monitoring capacity [182]. Countries with greater access to information gathering technologies (e.g., dense/accessible AIS coverage, established Passive Acoustic Monitoring (PAM) networks, repositories of satellite imagery or the capacity to task satellite time, and long-term, funded, research programmes) are better equipped to generate empirical evidence, whereas regions with limited resources or a lack of established research infrastructure may struggle to implement comprehensive monitoring programs, leading to significant knowledge gaps concerning local cetacean populations and the potential impacts of maritime vessels [183–185]. Areas with greater evidence (e.g., North American/European waters) are also where management tools, such as those implemented by the International Maritime Organization (IMO), including traffic separation schemes and areas to be avoided, or by local port and harbour authorities, such as seasonal or dynamic vessel speed regulations, and emissions/noise controls, are most frequently trialled or implemented [186–188]. In addition, the presence of regulatory frameworks in high-income countries (such as EU Habitats and Birds Directives, the Natura 2000 network or the U.S. Marine Mammal Protection Act) have directly stimulated investment in cetacean monitoring and research [189–192], whereas in regions of the global South, weaker implementation capacity and competing development priorities have limited sustained cetacean research despite legal protections [193]. All these differences could create a feedback loop where management policies and regulatory frameworks reflect monitoring capacity rather than true need based on impact risk, underscoring the importance of international cooperation for targeted capacity building and technology transfer [194–196]. Lastly, there may be other safety obstacles (such as piracy) that could hinder research and possibly affect the number of other maritime vessels on the water, especially in the global South [197,198].

It's finally important to note that in this systematic map a "central location" was chosen to indicate where the research has been conducted, as often the information on the full extent of the study area was missing or unclear, and it could vary the different types of impacts being studied. While this standardised approach allowed to place a study within the respective marine region, it does not allow to discern finer scale patterns within these regions or where the exact study locations are in relation to high vessel traffic areas.

## Cetacean species

The disparity in records between groups and individual species may be explained by a variety of factors including but not limited to: the number of species associated with each group, their geographic range and habitat preferences, their ecology, their sensitivity to particular vessel threats and their overall conservation status. To start, species with a cosmopolitan distribution (e.g., humpback whales, bottlenose dolphins) are more likely to be the subject of multiple studies on similar topics across their whole range, as opposed to species found in unique areas (e.g., finless porpoises) [199,200]. Species exhibiting preferences for coastal habitats and/or shallower waters are generally easier and less costly to study than those in offshore waters [201]. As a result, they are more likely to generate a higher number of records, a trend that may be further amplified by the typically high levels of vessel traffic in these areas, particularly near port and harbour entrances. Several species that inhabit heavy traffic coastal areas have impacts from vessels recognized as a primary threat to their conservation status, like in the case of North Atlantic Right whales [202], St. Lawrence Estuary belugas [203], Southern Resident killer whales [204], Atlantic humpback dolphins [205] or Hector's dolphins [206].

In addition to species' geographical distribution and/or migratory behaviour, vertical movement can also influence the amount of available evidence. Species with deep-diving habits, often coupled with cryptic behaviour such as avoiding boats and spending minimal time at the surface, are generally harder to encounter and study, as exemplified by beaked whales [207]. Conservation status will also likely influence the amount of effort dedicated to understanding a species' or population's threats and its survival, though it appears that charisma and accessibility are stronger driving factors [208]. For many species (especially porpoises and beaked whales) the main threats identified in the IUCN Red List are either noise-related, which is often studied through predictive approaches, or fall outside the inclusion criteria of this systematic map because they are associated with human activities rather than vessels themselves (e.g., coastal development, bycatch, plastic ingestion, or noise from active sonar; [209–214]).

Lastly, it is important to acknowledge that, in using record counts as a measure of research effort, the values presented could be influenced by long-term studies or establishment of research programmes which can result in multiple publications/records being generated. While the size of the marine regions used to collate records in this systematic map does not allow us to evaluate the extent of fine-scale spatial patterns in research efforts, some of the geographical differences that were, identified for individual species may be, at least in part, also related to the presence of such long-term studies or established research programmes. For example, nearly half (15/32) of the European studies on Atlantic bottlenose dolphins (Tursiops truncatus) were documented within UK waters. Similarly, virtually all of the identified records for killer whales (22/24) were collected in North American waters, there were no records available for Antarctic, African, European and Asian waters, despite this species having a global range. Of these North American records, majority focus on Southern Resident killer whales (e.g., [215–218]), while less research has focused on other populations, such as Northern Resident or transient killer whales (e.g., [219,220]). This suggests that research effort may not be evenly distributed between populations, not just species, as high-lighted by another study that considered evidence of vessel impacts for specific species subpopulations in the Arctic [71]. Because cetacean populations differ in distribution, habitat use, demography, and baseline status, the same vessel pressure may present very different risk(s) across sub-populations [221,222]. Spatially averaged or species-wide assessments ignore local hotspots of high overlap, place-specific sensitivities (e.g., small, closed, or breeding populations), demographic differences in response, and variation in traffic type and seasonality [222–227]. As such, these findings suggest that just because there appears to be a relatively high volume of available evidence for a species, this shouldn't be misinterpreted, or preclude from further efforts to investigate possible variability in exposure and responses at the population or sub-population level.

## Vessels

This map has shown that whilst interaction with commercial cargo ships and passenger vessels is recognised as a potential threat related to cetacean conservation [134,228], there is less empirical evidence than expected documenting their impacts [71], with only 41 out of 100 records providing empirical evidence. This gap is especially critical from a management perspective, as these are the target vessel categories of many regulatory frameworks, from the international IMO guidelines to national regulations, such as traffic separation schemes, speed-reduction measures and routeing changes [32,229].

For other vessel types, the relatively fewer records may reflect the fact that the focus of the study has been on the activity being performed, rather than on interactions between cetaceans and the vessels themselves. For instance, while fishing gear and of entanglement are widely recognised as main threats to cetaceans and much effort has been put into understanding and mitigating their impacts [230–232], this review identified fishing boats as some of the least represented vessel types. This however does not automatically translate in a lower risk associated with fishing boats, as the available records provide evidence of collisions with cetaceans as well as behavioural responses and changes in vocalizations [233–235].

Similarly, there is also limited available evidence regarding the potential impacts of motorized sailing boats (only one document, [236]). While this could be attributed to the fact that they operate mostly in coastal areas, many cetaceans (across most, if not all, families) also inhabit the same waters, making these vessels a potential source of impact. Instead, the number may be explained by differences in classification: as many sailing vessels use engine propulsion for large portions of their operational time [237], they may have been reported as "motorboats" in some of the analysed documents. It may also be possible that majority of the reports of the impacts of sailing vessels (and sailing competitions) may exist in other forms of grey literature (e.g., blog posts, videos) that were not included in this systematic map. This type of evidence can be imperfect and is generally not included in scientific reviews. However, it is still valuable and should be considered when it makes up the majority (if not all) of the available information.

The wide physical, operational, and geographical variability of maritime vessels complicates their classification, influencing both the type and severity of potential impacts on cetaceans and the availability of data [238,239]. Consequently, categorising vessels in this review involves important caveats that should be considered when interpreting the results. In particular, it is important to note that, for this systematic map, vessels were classified mainly based on their speed, size, distance to shore and whether they directly target cetaceans (e.g., eco-tourism boats) or are more likely to encounter them incidentally (e.g., fishing vessels). Therefore, some of the types (e.g., eco-tourism, cetacean research or fishing vessels, ferries) include vessels of a wide range of size and therefore a variety of potential impacts, that differ either in type or intensity [75]. Even seemingly straightforward terms such as "AIS vessels" or "AIS data" actually refer to a wide variety of vessel types, with categorization also being subject to user error as category is programme by the vessel operators instead of automatically assigned. In addition, geographical area can affect which vessel types are represented in AIS data. For example, in international waters, these likely include only class A vessels (those obligated to operate an AIS transponder under the International Maritime Organisation (IMO) Safety of Life at Sea (SOLAS) Regulation V/19 of 2004), whose data is also given priority on the satellite AIS receivers. In national waters however such assumptions cannot be made. While priority might still be given to class A vessels, the percentage of class B vessels is likely higher than in international waters and, since requirements for class B are set by the individual countries and can differ between EEZs (e.g., mandatory for commercial fishing vessels in USA but not Canada; [240,241]), generalizations in national waters cannot be made. In addition, vessels were assigned to a category other than "unspecified" only if the vessel type was explicitly stated in the publication. As such, when terms like "shipping" were found, records were considered as indicating "large unspecified vessels" since the term, while widely used to refer to commercial cargo traffic, does not inherently define a precise vessel type.

One of the notable challenges with this review was classifying the records into different vessel types, as very often (in almost 40% of cases) studies did not include specific details on the vessels included in the research. In many studies authors often referred to the stressor as simply "vessels" and no further information on vessel characteristics could be inferred from the text. This was most common for those studies considering the effects of underwater noise, when the source vessel was not specifically identified or cumulative noise exposure was being evaluated. Similarly, reports of stranding events or injuries on the animals were rarely able to explicitly state the size and type of vessel involved, because it was often unknown. There were cases, however, where the terminology used or the location of the study area allowed us to make informed decisions about the size of these "unspecified vessels". For example, documents referring to "commercial traffic" or "shipping traffic", or those evaluating noise inside shipping lanes were categorised as "likely-large unspecified vessels", as they would likely target large cargo or passenger vessels but didn't explicitly state so. Similarly, studies conducted in coastal waters (3–5 nautical miles from the coast) and away from main ports were assigned to the "likely-small unspecified vessels" category, as they probably referred to smaller recreational boats. While these decisions were discussed between the authors in an effort to ensure consistency, we acknowledge that the use of expert judgement introduces a level of subjectivity in the classification and that many of these studies including "unspecified vessels" could still refer to vessels of various sizes and types.

Inaccurate classification of vessel types limits our understanding of the threats they pose, potentially leading to over- or underestimation of impacts. Without a holistic view of how impacts vary by vessel type, informed decisions on management and risk mitigation are difficult. Therefore, it is essential to report vessel characteristics in detail whenever possible to achieve a comprehensive and nuanced understanding of their potential effects.

Despite the challenges in identifying and categorising vessel types involved in some records, this review can highlight specific knowledge patterns. In contrast to the lack of empirical evidence on fishing, sailing and commercial cargo and passenger vessels, there are records to suggest the presence of potential impacts associated with cetacean research activity itself, though effort appears to have focused mostly of dolphin species (Delphinidae and Monodontidae). While the recent years have seen a rapid growth in the use of remote sensing and autonomous technologies to quantify and

mitigate vessel impacts on cetaceans [242–244], as long as boat-based research will be conducted it would still be valuable to assess the potential impact of the research platform itself, especially on those species groups for which evidence is currently lacking. It is also true that unmanned aerial vehicles (UAVs), autonomous surface and underwater vehicles, and satellite remote sensing are increasingly being combined with PAM and biologging to map cetacean distribution, behaviour, and exposure to shipping noise and collision risk, with higher resolution and at lower cost than traditional ship-based surveys, thereby providing an emerging technological toolbox for evidence-based vessel-impact assessment and mitigation [245–247]. For example, passive acoustic monitoring (PAM) has moved beyond traditional moored recorders to include cabled observatories, seafloor telecommunication cables, and mobile platforms (such as underwater gliders, wave gliders, and other unmanned surface vehicles), greatly expanding spatial and temporal coverage of ship noise and whale occurrence [247,248]. These mobile PAM systems are also now being utilised in dynamic management schemes, where near-real-time detections of endangered North Atlantic right whales from gliders trigger mandatory vessel slow-downs and other risk-reduction measures over large regions and across seasons [249,250]. In parallel, satellite-linked telemetry of whales is increasingly combined with terrestrial and satellite AIS data on vessel movements to quantify spatial overlap, identify critical habitats and migratory corridors that intersect high-traffic routes, and evaluate ship-strike risk at regional and basin scales [244,251,252].

Lastly, it is important to note that the type of impacts considered in relation to each vessel type requires further scrutiny. Research effort might not be equally distributed across impacts, and indeed we noted that, for example, effects of behavioural disturbance from recreational motorboats appear to be more studied [253–257] than those of noise [145,258]. Importantly, this systematic map does not analyse or consider different types of impacts separately or based on their severity, and as such, record frequency cannot, and should not be interpreted as a proxy for the scale or severity of the threat that a vessel type might pose to a particular species.

Beyond individual impacts there is a growing body of evidence pointing to the necessity of considering the cumulative impacts that vessels may have on cetaceans, when investigating the conservation concerns that these stressors may have [259–262]. Approaches that attempt to take a more holistic view of impacts, such as Cumulative Impact Assessments (CIAs) and Ecosystem-Based Models (EBMs), can provide a more inclusive framework for trying to evaluate the combined effects of diverse maritime activities on marine mammal populations [263,264]. However, attempting to evaluate multiple stressors and their interactions over space and time to provide a more realistic understanding of anthropogenic pressures is something that is widely pointed to as being one of the biggest challenges when managing marine systems [265,266]. This comprehensive perspective is essential for informing policy development and it underscores the urgent need for standardized methodologies in impact assessment and monitoring [261,267]. Indeed, few of the documents identified from this review attempted to consider multiple vessel types (e.g., [51,257,268,269]) and none explicitly assessed the cumulative impacts from all those vessels. Similarly, studies that do attempt to evaluate cumulative exposure tend to focus on a single type of stressor (e.g., noise) and often examine different sources in addition to vessels, rather than multiple vessel types [270–272], while impacts from different stressors are understudied and empirical evidence is especially lacking [273].

## Future considerations

The volume of evidence analysed in this systematic map supports our understanding that vessels, regardless of their type, pose a range of ubiquitous threats to all cetaceans. Yet in order to design effective measures to ensure that vessels do not pose a conservation risk, more empirical evidence on the threats associated with different vessel types is required. Therefore vessel types and their attributes should be clearly documented in every study to ensure that management tools (such as speed-reduction zones, shipping lane adjustments, dynamic management areas, noise management [32,229,274,275]) are appropriate and proportionate to effectively mitigate the potential impacts of vessels while minimising unnecessary disruption to maritime traffic where possible.

Recognizing the differences in species-specific responses and vessel characteristics is critical for tailoring conservation policies and monitoring strategies effectively. Different cetacean species exhibit varied sensitivities to vessel disturbances [256,276,277], and vessel types differ substantially in their size, speed, noise frequency, and operational patterns, which may result in different impacts [278,279]. This heterogeneity necessitates conservation approaches that are nuanced and adaptive, ensuring that mitigation measures are both species-appropriate and vessel-specific. As such, future field studies should be dedicated to those under-represented species (like most porpoise and all beaked whale species) and vessel types (such as fishing, sailing or seismic vessels), making sure to include, where possible, clearer details about the vessels that are being considered, like information on engine size, overall length, speed and hull type. Efforts should also focus on Asian, African, South American, international and polar waters, where evidence is still lacking. With the increase in maritime activities and the expansion to newer, previously inaccessible parts of the ocean, it is also important to extend research efforts to offshore areas and remote regions like the Arctic and Antarctic, where populations have been less subjected (and therefore more sensitive) to anthropogenic pressures [41,280,281]. While these regions are inherently more challenging to survey, coordinated international effort should employ recent technological innovations (such as PAM, high-resolution satellite imagery, or unoccupied aerial systems) to expand data coverage to now provide the opportunity to capture evidence even in such remote areas and cryptic species [243,282]. These technological advances also provide opportunities for continuous, non-invasive, long-term monitoring data that was previously unattainable, for cetaceans and vessels alike.

It is however important for this new research to follow more standardised protocols. The heterogeneity in methodologies identified across studies highlights the need for greater standardisation in data collection, reporting and vessel classification. While for some stressors (e.g., noise) vessel types have been well characterised and responses to different vessel types are well studied [283], such level of detail may not be available for other stressors. Only recently, risk of lethal collision with large whales has been described for vessels of different sizes, while previously it was solely based on vessel speed [238]. And for other impacts, like behavioural disturbance due to vessel presence, variation in responses to different vessel types have not been analysed yet and the vessel categorization systems used for noise (by type and operational characteristic) and collision (by size) may not be suitable to assess this impact as a variable relating to activity might need to be incorporated to account for those vessels that are more likely to encounter cetaceans due to their operations (e.g., whale-watching and fishing vessels) or those that have the ability to actively pursue the animals (e.g., whale-watching or recreational boats, as opposed to commercial vessels with stricter routes and schedules). Despite these challenges, developing shared protocols for describing vessel characteristics and defining vessel categories according to size and activity, and developing consistent monitoring approaches would enhance comparability across studies and regions, improve the transferability of results to management contexts, and facilitate the integration of evidence into policy and regulatory frameworks.

In addition to guiding new empirical research, this systematic map can also guide future systematic evidence analysis. Firstly, future synthesis should account for different responses from the animals, with clear distinctions between behavioural, social or physiological responses, vocalization changes, and lethal or sublethal injuries, as well as short- or long-term consequences. Secondly, future reviews should explore seasonal patterns in the available evidence and the possible lack of data during months where survey conditions are commonly unfavourable (like boreal winter, austral summer, monsoon seasons). High-resolution spatial patterns should also be investigated to assess possible biases at a local scale and between inshore and offshore waters, which necessitates recording the full extent of the study area rather than a central location. Thirdly, with the emergence of new technologies, future synthesis could include information of the data sources used for both cetaceans (e.g., land- or ship-based studies, visual or acoustic, eDNA) and vessels (e.g., land- or boat-based surveys, AIS data), with particular interest on temporal and spatial patterns. It will also be important to compare distribution of empirical and predictive evidence, as that is likely to differ between vessel types, species (or species groups) and response types. For example, research using AIS data might be more predictive in nature, and studies on

elusive species might rely more strongly on modelling due to the lack of observational records. At a smaller scale, future systematic literature reviews should focus on assessing potential knowledge gaps at the population and sub-population level for those apparently better-studied species, like humpback whales, bottlenose dolphins, killer whales and sperm whales. For this to happen, a higher spatial resolution than the one used in the current study is needed. In particular, extracting information of the size on the study area (rather than the centre location) would allow to better evaluate overlap with both high vessel traffic zones and protected areas, to assess whether cetaceans are being studied and protected where they are actually more vulnerable. In addition to systematic literature reviews, future studies focusing on data-deficient species or vessel types should also attempt to collate information for any types of documentation and media (including generally disregarded sources like blog posts or videos) as well as conducting interviews with marine users, like has been done to document entanglement and bycatch for example [284–288].

Lastly, while this systematic map cannot result in direct suggestions on managing vessel traffic, it provides an established, user-friendly evidence base that is easier to interrogate, as opposed to reading each singular document identified in this review. As such, policy makers and managers, both at the local and international level, now have the means to explore the collated resources and form the basis for developing evidence-based guidance and regulations regarding the impacts on vessels. The outcomes of this systematic map can also support international mitigation efforts that emphasise data acquisition and analysis, potentially securing targeted funding toward understudied species, regions, or vessel types. Additionally, it highlights the need for harmonised reporting, which could promote international coordination and compliance and the formulation of industry bodies' best-practice guidelines.

## 5. Conclusion

In a world of increasing maritime activity, in particular in the form of vessel traffic, this systematic map summarises the distribution of available empirical studies on the impacts of maritime vessels on cetacean species. It highlights that our current understanding is built on the responses of few species in specific parts of the world. Future research should therefore seek to build an understanding related to the currently data-deficient species or species groups (such as porpoises and beaked whales) and marine regions (such as African, polar and international waters). Similarly, not all vessel types have received equal research attention, and for some vessel types the evidence remains predominantly of predictive rather than empirical nature. In addition, the ambiguity on vessel characteristics due to the quality of information provided in the document can lead to further uncertainty. Effort should be made to strive for more detailed records of the vessels involved in the studies and to assess the impacts of different vessels on different species, both singularly and cumulatively.

Finally, recognizing and addressing the global disparities in monitoring capacity through capacity-building initiatives, technology transfer, and collaborative research networks is essential for equitable conservation outcomes. By fostering partnerships among developed and developing regions, sharing expertise, and promoting access to cutting-edge monitoring technologies, the global community can work towards a more unified and effective approach to marine mammal conservation. This comprehensive, multi-faceted strategy is vital for safeguarding marine biodiversity in an era marked by escalating maritime pressures and rapid environmental change.

## Supporting information

**S1 Appendix. Additional figures and tables presenting spatial and temporal patterns for empirical and predictive records, as well as the number of records available on each species for each vessel type.**
(DOCX)

**S1 Data. Excel file containing the information extracted from the documents included in this systematic map.**
(XLSX)

## Acknowledgments

We would like to thank Chris Reilly for their contribution as secondary reviewer during title-and-abstract screening stage of this systematic map.

## Author contributions

**Conceptualization:** Valeria Ferrari, Emily Hague, Lauren McWhinnie.

**Data curation:** Valeria Ferrari.

**Formal analysis:** Valeria Ferrari.

**Investigation:** Valeria Ferrari.

**Methodology:** Valeria Ferrari, Emily Hague, Marija Sciberras, Lauren McWhinnie.

**Supervision:** Karen A. Alexander, Patrick D. O'Hara, Lauren McWhinnie.

**Visualization:** Valeria Ferrari.

**Writing – original draft:** Valeria Ferrari, Lauren McWhinnie.

**Writing – review & editing:** Valeria Ferrari, Emily Hague, Marija Sciberras, Karen A. Alexander, Patrick D. O'Hara, Lauren McWhinnie.

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
