## [Decision Letter · Decision Letter 0]

14 Jan 2026

PONE-D-25-59943Understanding the global threats on cetaceans from maritime vessels: a systematic mapPLOS One

Dear Dr. Ferrari,

Thank you for submitting your manuscript to PLOS ONE. After careful consideration, we feel that it has merit but does not fully meet PLOS ONE’s publication criteria as it currently stands. Therefore, we invite you to submit a revised version of the manuscript that addresses the points raised during the review process.

If applicable, we recommend that you deposit your laboratory protocols in protocols.io to enhance the reproducibility of your results. Protocols.io assigns your protocol its own identifier (DOI) so that it can be cited independently in the future. For instructions see: https://journals.plos.org/plosone/s/submission-guidelines#loc-laboratory-protocols. Additionally, PLOS ONE offers an option for publishing peer-reviewed Lab Protocol articles, which describe protocols hosted on protocols.io. Read more information on sharing protocols at . Additionally, PLOS ONE offers an option for publishing peer-reviewed Lab Protocol articles, which describe protocols hosted on protocols.io. Read more information on sharing protocols at https://plos.org/protocols?utm_medium=editorial-email&utm_source=authorletters&utm_campaign=protocols..

We look forward to receiving your revised manuscript.

Kind regards,

Vitor Hugo Rodrigues Paiva, Ph.D.

Academic Editor

PLOS One

Journal Requirements:

2. We note that your Data Availability Statement is currently as follows: All relevant data are within the manuscript and in Supporting Information files.

3. We note that Figure 2 in your submission contain map images which may be copyrighted. All PLOS content is published under the Creative Commons Attribution License (CC BY 4.0), which means that the manuscript, images, and Supporting Information files will be freely available online, and any third party is permitted to access, download, copy, distribute, and use these materials in any way, even commercially, with proper attribution. For these reasons, we cannot publish previously copyrighted maps or satellite images created using proprietary data, such as Google software (Google Maps, Street View, and Earth). For more information, see our copyright guidelines: http://journals.plos.org/plosone/s/licenses-and-copyright.

1. You may seek permission from the original copyright holder of Figure 2 to publish the content specifically under the CC BY 4.0 license.

4. Please include a caption for figure 4,5 and 6.

Reviewers' comments:

Reviewer's Responses to Questions

**Comments to the Author**

1. Is the manuscript technically sound, and do the data support the conclusions?

Reviewer #1: Yes

Reviewer #2: Partly

2. Has the statistical analysis been performed appropriately and rigorously? 

Reviewer #1: Yes

Reviewer #2: No

3. Have the authors made all data underlying the findings in their manuscript fully available?

Reviewer #1: Yes

Reviewer #2: Yes

4. Is the manuscript presented in an intelligible fashion and written in standard English?

Reviewer #1: Yes

Reviewer #2: Yes

5. Review Comments to the Author

Reviewer #1: This manuscript addresses an important and timely topic: the impacts of maritime traffic on cetaceans. The authors apply a systematic map methodology to synthesize an impressive body of literature and identify knowledge gaps across species, vessel types, and locations. The topic is highly relevant to marine conservation, maritime spatial planning, and environmental management. The dataset will be a useful resource for researchers and managers. The manuscript is generally well structured, methodologically transparent and clearly written. The methods applied in the review is globally sound and the effort to synthesize predictive tools across studies is appreciated.

However, the manuscript currently suffers from significant weaknesses:

• in the Introduction with the use of many outdated or inappropriate references, as well as important omissions in the literature,

• In the Results, several areas require clarification, correction with taxonomic inconsistencies and errors in figures

• In the Discussion where strengthening, or deeper critical interpretation are needed. Some methodological choices need justification and the discussion could more explicitly connect findings to management implications, biases in the literature, and future research priorities.

Finally, the manuscript sometimes reads more like a descriptive catalogue than a fully analytical synthesis, in particular the discussion. These issues substantially weaken the scientific robustness and global relevance of the paper. Major revisions are therefore required before the manuscript can be considered for publication.

Major Comments

The manuscript claims to focus on “direct impacts from maritime vessels themselves” but the boundary between “direct” and “secondary” impacts is not always clear. For example, noise from vessels is included but noise from vessel-associated activities (e.g., pile driving) is excluded. This distinction can be justified and the rationale needs to be more explicit.

The introduction could better articulate why a systematic map (rather than a systematic review or meta-analysis) is the appropriate tool: what specific decisions or management needs does a map support?

The manuscript acknowledges that more than 40% of records could not be assigned a vessel type. For unspecified vessels, “expert judgement” was used to infer vessel size/type. This introduces non-trivial classification bias, particularly given the high proportion of records lacking vessel-type attribution and and the central role of vessel type in the presented results. This is a major limitation and the authors do not sufficiently interrogate why this occurred: was it due to poor reporting in primary studies or limitations in the classification scheme? Moreover, the use of expert judgement to infer vessel type when not explicitly stated introduces subjectivity: How was consistency ensured? Was a second reviewer consulted for ambiguous cases?

The manuscript completely omits the impact of sailing boat competitions (e.g. Vendée Globe, The Ocean Race), despite their global footprint, the known collisions and operation at high speeds in ecologically sensitive offshore regions. There are multiple mentions of collisions (sometimes with videos where we can identify the species). Data on these events and vessel trajectories do exist and their exclusion leaves an important gap in the assessment of vessel-related pressures. At minimum, this limitation should be acknowledged and justified.

In the results section, the authors do not explore publication bias, research capacity disparities or the influence of regulatory frameworks. The distinction between empirical and predictive studies is useful but the manuscript does not analyze how these two evidence types complement or contradict each other. The use of record counts may disproportionately reflect heavily studied systems or species. In particular, a single long-term study contributing multiple records could inflate the perceived strength of the evidence. This issue is not explicitly or adequately addressed.

All impacts are treated equally (e.g. brief behavioral response vs lethal strike). This is acceptable for a map but care should be given to avoid misinterpretation of record frequency as impact severity.

There is no discussion on how spatial assignment may influence results: using the “centre of the study area” may distort representation for large study regions (e.g., migratory corridors, ocean basins). The authors should discuss potential spatial bias introduced by this simplification.

The discussion needs to be expanded and strenghtened. As such, it does not sufficiently connect findings to:

• management needs (e.g., IMO guidelines, MPAs, vessel speed regulations),

• technological advances (AIS, PAM, satellite tracking),

• global heterogeneity in monitoring capacity.

The authors note that vessel heterogeneity is poorly captured in the literature but do not propose concrete solutions (e.g., standardized reporting frameworks, integration with AIS metadata).

While mentioned, cumulative impacts (e.g., traffic density + noises) are not fully explored.

Finally, the manuscript could better articulate how this systematic map can guide:

• future systematic reviews,

• targeted field studies,

• policies.

In its current version, the discussion overlooks major opportunities to strengthen the relevance and impact of this work, particularly by failing to articulate implications for conservation policy, monitoring strategies and methodological standardization.

Detailed Comments

Ensure consistent terminology (e.g., “vessel traffic,” “maritime vessels,” “shipping”).

Given that this manuscript presents itself as a systematic review, the presence of outdated, irrelevant and missing references in the introduction is particularly concerning and undermines the credibility of the evidence synthesis.

Here is a non exhaustive list of the citation flaws of the introduction:

Citations [2,3] do not match the subject they are cited for “For the past few decades, marine

transportation has undergone continuous increasing growth in terms of the number of vessels at sea” and they are also too “old” (2012, 2016), please find better matching and more recent papers

Same for citations 4,5

The citation for the following is needed: “International Chamber of Shipping documented that commercial shipping (at least 100 gross tons) now accounts for 105,500 vessels, amounting to around 85% of trade occurring globally”

The citation for “new shipping routes are being discussed” (7) dates back 2017, an updated document on these discussions would be appreciated.

Citations 9,10 are too old, more recent references are needed here.

Citations 11 and 12 are outdated (24 years-old!!).

A major pressure for the marine megafauna is missing here (citation 14): the risk of ship strike. “This growth in maritime activity has also resulted in an increased awareness of the resultant pressures vessels can place on marine environments including various forms of noise, air and water pollution”. Here it is necessary to cite at least one paper per ocean basin: Atlantic, Pacific, Mediterranean, Indian…

e.g. 1- for the Caribbean which is a region completely missed in this paper:

Bénédicte Madon, Damien Le Guyader, Jean-Luc Jung, Benjamin De Montgolfier, Pascal Jean Lopez, Eric Foulquier, Laurent Bouveret, Iwan Le Berre,

Pairing AIS data and underwater topography to assess maritime traffic pressures on cetaceans: Case study in the Guadeloupean waters of the Agoa sanctuary,

Marine Policy, Volume 143, 2022, doi.org/10.1016/j.marpol.2022.105160.

2- for the Med: Maxime Sèbe, Léa David, Frank Dhermain, Sophie Gourguet, Bénédicte Madon, Denis Ody, Simone Panigada, Hélène Peltier, Linwood Pendleton,

Estimating the impact of ship strikes on the Mediterranean fin whale subpopulation,

Ocean & Coastal Management, Volume 237, doi.org/10.1016/j.ocecoaman.2023.106485.

Citation 15 is again outdated, a more recent reference should be added.

Citation 18 does not match the subject it is quoted for, find another reference.

A citation to avoid citing only papers in the Pacific (with citations 21-22 and/or with the new group in place of citation 14):

Fumagalli, M., Cesario, A., Costa, M. (2019). Where Dolphins Sleep: Resting Areas in the Red Sea. In: Rasul, N., Stewart, I. (eds) Oceanographic and Biological Aspects of the Red Sea. Springer Oceanography. Springer, Cham. https://doi.org/10.1007/978-3-319-99417-8_17

Citation 26 is not relevant where it is cited. More citations with more varied locations that just the Pacific West is required for vessel strikes in: “A broad range of threats have been connected to maritime traffic, and subsequently documented for many marine species but cetaceans as a group have been identified as being particularly vulnerable to direct threats from the vessels themselves such as: underwater noise [21,22], discharges and contaminants [23–25], and vessel strikes [26–28].

It is surprising that the authors did not find any paper on the issue of vessel strike and the emblematic North Atlantic right whales!

I do not agree with this: “While some impacts (like collisions between cetaceans and vessels) are limited to the exact location of the vessels [45],”, the wording as is now, confuses between where the event happens vs where the ecological impact manifests, please rewrite to clarify.

In the group 58-61: this reference should be added: Schaffar A, Madon B, Garrigue C, Constantine R (2013) Behavioural effects of whale-watching activities on an Endangered population of humpback whales wintering in New Caledonia. Endang Species Res 19:245-254 https://doi.org/10.3354/esr00466

It is again surprising that in the group of citations 82-90, there is none on work on North Atlantic right whales.

For “Recent advances in technologies such as passive 121 acoustic monitoring (PAM) and remote sensing (“Automatic Identification Systems (AIS) data) have advanced our understanding and ability to evaluate and in some cases mitigate the threats posed by vessels” previous proposed citation should be used as “e.g.”: Bénédicte Madon, Damien Le Guyader, Jean-Luc Jung, Benjamin De Montgolfier, Pascal Jean Lopez, Eric Foulquier, Laurent Bouveret, Iwan Le Berre, Pairing AIS data and underwater topography to assess maritime traffic pressures on cetaceans: Case study in the Guadeloupean waters of the Agoa sanctuary, Marine Policy, Volume 143, 2022, doi.org/10.1016/j.marpol.2022.105160.

In the results

Latin names of species are missing. Please do not cite only generic names but use Latin ones also.

Fig.5 to 8: The classification in the figure lacks consistency, as it juxtaposes taxonomic ranks from different hierarchical levels—such as the suborder Mysticeti, the superfamily Physeteroidea, and families like Phocoenidae and Ziphiidae—alongside the informal grouping 'Delphinidan'. This undermines taxonomic coherence and comparability.

Fig.9 is wrong. Where are vessel types?

S5 Fig: Annual number of predictive records included in this systematic review, with focus on the document type. Without years 1981-1992, should be redone removing years 1981 to 1992.

Figures could be more interpretive—e.g., maps showing density of studies relative to global shipping intensity.

Reviewer #2: The research and referencing work is impressive. The classification methodology is equally impressive.

There is some confusion between the title of the article: Understanding the global threats to cetaceans from maritime vessels: a systematic map, and the discussion within it.

Indeed, when reading the title, one expects a compilation of impacts over time, by species or groups of species, type of vessel, etc.

The discussion presents the number of references for each, but no summary of existing data.

The title led me to expect tables and maps showing impacts, target species, sources, and their evolution over time. In the end, it is a library summary rather than a scientific review. I would expect a review to summarize existing data and conclusions, not just list available publications.

l.65 : Precise the environmental impacts. As the authors go on to detail for cetaceans, just mention the impacts here.

l.73: Precise, in few words, the negative impact on cetaceans.

l.95: I will place the quote [66] after this sequence: “ranges as the vessel”

l.125: please put a comma after “by vessels”.

l.208: Why choose to group Physeteridae and Kogiidae together when their habitats are different and the latest publications on classification suggest differentiating between them?

Table 1: Have you made a distinction between traditional hulls and those using foils when it comes to sailing boats? Are there any usable studies?

l.419: please indicate the proportion. This will help the reader.

l.452: The use of new techniques (acoustics, eDNA) has also made it possible to study certain species. These techniques have emerged recently, or have been in use for less time than traditional monitoring methods involving direct observation. It would have been interesting to include the techniques used in the results, or even the publication dates associated with the techniques. The same applies to the chronology of the number of studies by species or groups of species.

L.496-497: But is there a direct impact from fishing boats? It would be interesting to clarify this here.

6. PLOS authors have the option to publish the peer review history of their article (what does this mean?). If published, this will include your full peer review and any attached files.). If published, this will include your full peer review and any attached files.

.

Reviewer #1: **Yes:** Bénédicte MadonBénédicte Madon

Reviewer #2: No

To ensure your figures meet our technical requirements, please review our figure guidelines: s://journals.plos.org/plosone/s/figures

You may also use PLOS’s free figure tool, NAAS, to help you prepare publication quality figures: s://journals.plos.org/plosone/s/figures#loc-tools-for-figure-preparation.

---

## [Author Response · Author response to Decision Letter 1]

27 Feb 2026

Thank you for taking the time to consider our manuscript for publication in PLOS ONE. We thank the reviewers and editor for taking the time to provide detailed feedback and considerate comments. We have now addressed all of the reviewers comments and have provided a summary of our responses in the "Response to Reviewers" document, as well as two copies of the Revised Manuscript, one unmarked and one with 'Tracked Changes'.

---

## [Decision Letter · Decision Letter 1]

16 Apr 2026

Exploring the evidence of direct threats to cetaceans from maritime vessels: a systematic map

PONE-D-25-59943R1

Dear Dr. Ferrari,

We’re pleased to inform you that your manuscript has been judged scientifically suitable for publication and will be formally accepted for publication once it meets all outstanding technical requirements.

An invoice will be generated when your article is formally accepted. Please note, if your institution has a publishing partnership with PLOS and your article meets the relevant criteria, all or part of your publication costs will be covered. Please make sure your user information is up-to-date by logging into Editorial Manager at Editorial Manager® and clicking the ‘Update My Information' link at the top of the page. For questions related to billing, please contact  and clicking the ‘Update My Information' link at the top of the page. For questions related to billing, please contact billing support..

Kind regards,

Vitor Hugo Rodrigues Paiva, Ph.D.

Academic Editor

PLOS One

Additional Editor Comments (optional):

Reviewers' comments:

Reviewer's Responses to Questions

**Comments to the Author**

1. If the authors have adequately addressed your comments raised in a previous round of review and you feel that this manuscript is now acceptable for publication, you may indicate that here to bypass the “Comments to the Author” section, enter your conflict of interest statement in the “Confidential to Editor” section, and submit your "Accept" recommendation.

Reviewer #1: All comments have been addressed

2. Is the manuscript technically sound, and do the data support the conclusions?

Reviewer #1: Yes

3. Has the statistical analysis been performed appropriately and rigorously? 

Reviewer #1: N/A

4. Have the authors made all data underlying the findings in their manuscript fully available?

Reviewer #1: Yes

5. Is the manuscript presented in an intelligible fashion and written in standard English?

Reviewer #1: Yes

6. Review Comments to the Author

Reviewer #1: (No Response)

7. PLOS authors have the option to publish the peer review history of their article (what does this mean?). If published, this will include your full peer review and any attached files.). If published, this will include your full peer review and any attached files.

.

Reviewer #1: **Yes:** Bénédicte MadonBénédicte Madon

---

## [Editor Report · Acceptance letter]

PONE-D-25-59943R1

PLOS One

Dear Dr. Ferrari,

I'm pleased to inform you that your manuscript has been deemed suitable for publication in PLOS One. Congratulations! Your manuscript is now being handed over to our production team.

Kind regards,

on behalf of

Dr. Vitor Hugo Rodrigues Paiva

Academic Editor

PLOS One